# A Compound Control Based on the Piezo-Actuated Stage with Bouc–Wen Model

**DOI:** 10.3390/mi10120861

**Published:** 2019-12-07

**Authors:** Jiwen Fang, Jia Wang, Chong Li, Wei Zhong, Zhili Long

**Affiliations:** 1School of Mechanical Engineering, Jiangsu University of Science and Technology, Zhenjiang 212003, China; wjjzhb@just.edu.cn (J.W.); lichong@just.edu.cn (C.L.); zhongwei@just.edu.cn (W.Z.); 2Department of Mechanical Engineering, Harbin Institute of Technology, Shenzhen 518055, China; longworking@163.com

**Keywords:** piezo-actuated stage, Bouc-Wen hysteresis model, feedforward compensator

## Abstract

The piezoelectric actuator (PA) is one of the most commonly used actuators in a micro-positioning stage. But its hysteresis non-linearity can cause error in the piezo-actuated stage. A modified Bouc–Wen model is presented in this paper to describe the hysteresis non-linearity of the piezo-actuated stage. This model can be divided into two categories according to the input frequency: rate-independent type and rate-dependent type. A particle swarm optimization method (PSO) is employed to identify these parameters of the Bouc–Wen hysteresis model. An inverse model feedforward compensator is established based on the modified Bouc–Wen model. The fuzzy proportional-integral-derivative (PID) controller combined with the feedforward compensator is implemented to the piezo-actuated stage. The experimental results indicate that the proposed control strategy can compensate for the hysteresis phenomenon.

## 1. Introduction

The piezoelectric actuator has been widely utilized in ultra-precision positioning. However, the existence of hysteresis of the piezoelectric ceramics can lead to poor control performance of the stage. Without loss of generality, there are two ways to alleviate the hysteresis effects of the piezoelectric actuator: one, an inverse control based on the hysteresis model is used to address the non-linearity of piezoelectric actuator (PA); two, the non-linearity of the piezoelectric actuator is taken as a disturbance, and the robust adaptive control without hysteresis model can handle the non-linearity by suppressing the disturbance [1]. To compensate for the effect of the hysteresis of the piezoelectric actuator, the common method is to implement a hysteresis model based on several control strategies. Various hysteresis models are used to describe the hysteresis behavior. Among these models based on the approximation of the hysteresis, the Bouc–Wen model is a common hysteresis model, and its differential equation is compact and efficient. Because of only computing one auxiliary non-linear differential equation, this model of the piezoelectric actuator has the advantage of computational simplicity [2]. Jinqiang Gan proposed a generalized Bouc–Wen hysteresis model by applying relaxation functions to describe rate-dependent and rate-independent hysteresis behaviors of piezoelectric actuator [3]. A modified Bouc–Wen model, identified by the particle swarm optimization (PSO) method, is employed to estimate the hysteresis of the piezo-actuated stage [4]. The feedforward control of an inverse multiplicative structure with the multivariable Bouc–Wen model can avoid additional computation [5]. Hybrid adaptive differential evolution and the Jaya algorithm are utilized to identify the parameters of the Bouc-Wen model of the PA [6].

For solving the problem of the hysteresis phenomenon of the piezoelectric actuator, the compensator can be broadly classified as follows: open-loop compensation control with an inverse hysteresis model [7], the closed-loop displacement control, and feedforward controller and closed-loop controller [8]. For the compensation of hysteresis nonlinearity, identifying these parameters of the hysteresis model is just the first step and the basis of a compound control. The compensator of the hysteresis has attracted many researchers. Li Yangmin proposes an inverse hysteresis compensator based on the rate-dependent Bouc–Wen model [8]. A compound control is designed by combining inverse model feedforward compensation control and feedback control. In this type of control, the hysteresis effect of the PA is eliminated by the inverse model feedforward compensation control, and the error, the dynamics and the disturbance of the system are suppressed by feedback control. The identified hysteresis will be employed to design the feedforward controller to compensate hysteresis of the piezo-actuated stage. A compound control strategy based an inverse hysteresis model and a proportional-integral-derivative (PID) feedback controller is employed for the compensation of positioning errors [9]. A non-linear feedforward compensation controller based on the Bouc–Wen model describing bias-rate-dependent hysteresis is combined with the displacement PID control to improve the displacement accuracy of the actuator [10].

The paper tries to design a Bouc–Wen model of the piezo-actuated stage by considering the input frequency. Firstly, the mechanical structure of the piezo-actuated stage is introduced in Section 2. Section 3 provides the parameters identification of Bouc–Wen hysteresis model using PSO. The compound control, feedforward control of an inverse hysteresis model combined with fuzzy PID closed-loop feedback control, is presented in Section 4. Section 5 describes the experimental results. Finally, the conclusion is given in Section 6.

## 2. Mechanical Structure of Piezo-Actuated Stage

The flexure hinges exhibit several advantages, such as mode friction, no lubrication, lightweight and zero backlash. Piezo-actuated compliant mechanisms based on flexure hinges have been widely utilized in precision engineering.

Circular flexure hinges can provide small in-plane deflection, so they can achieve a small range but high-precision positioning. The elliptical flexure hinges are suitable for better flexibility with longer fatigue life because of the low maximum stress. A leaf-spring flexure hinge is usually utilized to generate high precision positioning of a relatively large range due to its low bending stiffness [11]. Therefore, Circular flexure hinges are selected to transmit the motion for the piezo-actuated stage.

Combining the compliant mechanisms, the CAD model of the piezo-actuated stage including eight flexure hinges is shown in Figure 1. These circular flexure hinges are utilized to transmit the motion actuated by PA. Eight symmetrical flexible hinges form double parallel four-bar guide mechanism. This guide mechanism can extend in the motion direction without coupling displacement.

The aluminum alloy 7075-T6 is utilized for the section of the precision stage due to high admissible elastic strain. Its material parameters are stated as follows [12]: Young’s modulus *E* = 71 Gpa, density *ρ* = 2800 kg/m^3^, and Poisson’s ration *μ* = 0.33. The dimension parameters of flexure hinges are designed as follows: *R* = 3.5 mm, *t* = 3 mm, *b* = 14 mm, *L* = 10 mm. Consequently, we obtain the result of the guide mechanism: *k*_e_ = 47.52 N/μm. It satisfies this condition that the driven force is larger than the sum of the elastic restorative force and the inertial force. The calculation shows that the system is reasonable.

The model and motion schematic diagram of the piezo-actuated stage are illustrated in Figure 2.

The motion displacement of this guide mechanism is very small and at the micrometer level. But the length of the link based on two flexure hinges is a few millimeters. Therefore, the rotation angle of a single link based on two flexure hinges is estimated by:(1)$$\theta \;=\arctan\frac{\Delta x}{L}\;\approx \;\frac{\Delta x}{L}$$

According to the law of conservation of energy, the work generated by the driven force is equal to the elastic potential energy stored by 8 flexure hinges. Without considering the stiffness of the tensile direction, the equation is expressed as follows:(2)$$W\;=\;\frac{1}{2}F\Delta x\;=\;\frac{1}{2}k_{e}\Delta x^{2}\;=\;\frac{1}{2}k_{r}\theta^{2}\;\times \;8$$
where *k*_e_ is the stiffness of the guide mechanism in the motion direction, and *k*_r_ represents the rotation stiffness of the flexure hinge.

Under the operation of the moment about the *z*-axis, the rotation stiffness formula of right circular flexure hinge as a pure rotation is given by
(3)$$k_{r}\;=\;\frac{2Eb\sqrt[{2}]{t^{5}}}{9\pi \sqrt{R}}$$
where *E* is the Young’s modulus of the material of this stage, *b* is the depth of the flexure hinges, *t* the thinnest thickness of the right circular flexure hinge, and *R* is the radius of the circle.

## 3. Parameter Identification of the Modified Bouc–Wen Model by Using Modified Particle Swarm Optimization (PSO)

To reveal the hysteresis characteristics of this piezo-actuated stage, a set of voltage signals with different input frequencies is employed to stimulate this stage. The hysteresis curves of this stage shown in Figure 3 are obtained by utilizing several sine voltage signals in different frequencies from 0.5 to 10 Hz ($Vol\;=\;4.58\;+\;4.58\;*\;\sin(2\pi ft\;-\;\frac{\pi }{2})\;f\;=\;\;0.5,\;1,\;5,\;10$). The experimental setup used is described in Section 5. From Figure 3, it is seen that the hysteresis loop of this stage varies little in the low-frequency voltage input under 1 Hz. It is known that the increase of the input frequency will give rise to the increase in the width of the voltage-displacement curve but a decrease in the height of the curve [13]. Therefore, the hysteresis curve is rate-dependent when the input frequency is greater than 1 Hz. The hysteresis phenomenon states that the relationship between the input voltage and the output displacement manifests nonlinearity, multivalue and nonlocal memoryless [14,15,16]. Consequently, the hysteresis characteristic of the PA is generally divided into two parts according to the input frequency: one, the hysteresis phenomenon is rate-independent when the input frequency is under 1 Hz; two, if the input frequency is larger than 1 Hz, the hysteresis becomes serious and needs to be described by a rate-dependent model.

To deal with the hysteresis phenomenon, many methods have been proposed to identify the parameters of hysteresis models, such as PSO [4,8,17], direct integration method [18]. PSO is an optimization method suitable for global optimization. The algorithm takes inspiration from the regularity of the foraging behavior of the animal cluster. The global group movement is an evolution process to achieve the optimal solution by solving space problems from disorder to an orderly process. The main strong advantages of the PSO algorithm are easy and simple to implement, and it has few parameters that need to adjusted [19].

In PSO, each particle denotes a design point that has its own position and velocity. The *i*th particle position can be represented as follows:(4)$$X_{i}\;=\;(x_{i1},\;x_{i2},\;\cdots ,\;x_{i(n-1)},\;x_{in})$$

The velocity for the *i*th particle is expressed as follows:(5)$$V_{i}\;=\;(v_{i1},\;v_{i2},\;\cdots ,\;v_{i(n-1)},\;v_{in})$$

The inertia weight can promote excellent-searching by balancing the global search and the local search. The equation for updating the particle velocity and position based on a general PSO is as follows:(6)$$\begin{array}{l} V_{id}^{k+1}\;=\;\varphi \{V_{id}^{k}+c_{1}r_{1}\;(P_{id}^{k}\;-\;X_{id}^{k})\;+\;c_{2}r_{2}\;(P_{g}^{k}\;-\;X_{id}^{k})\;\}\\ X_{id}^{k+1}\;=\;V_{id}^{k+1}\;+\;X_{id}^{k} \end{array}$$
where $\varphi \;=\;\frac{2}{2\;-\;C\;-\;\sqrt{C^{2}\;-\;4C}}$ is compression factor; $C=\;c_{1}\;+\;c_{2},\;\mathrm{and}\;C\;>\;4$; $V_{i}\;\in \;[\;-\;V_{\max},\;V_{\max}]$ represents the velocity of the particle; *k* is the current iteration number. *c*_1_, *c*_2_ are the acceleration factors, respectively. Both of the coefficients *c*_1_ and *c*_2_ are set to 2.1. *r*_1_, *r*_2_ ∈ [0,1] are independent random numbers; $P_{id}^{k}$ is the best current fitness of the *i*th particle in the *k*th iteration; $P_{g}^{k}$ is the best fitness of the whole particle swarm.

The formulas for updating the particle velocity and position based on the modified PSO is described as:(7)$$\begin{array}{l} V_{id}^{k+1}\;=\;\varphi \;\{\;wV_{id}^{k}\;+\;c_{1}r_{1}\;(P_{id}^{k}\;-\;X_{id}^{k})\;+\;c_{2}r_{2}\;(P_{g}^{k}\;-\;X_{id}^{k})\;\}\\ X_{id}^{k+1}\;=\;V_{id}^{k+1}\;+\;X_{id}^{k} \end{array}$$
where *w*, playing an important role in balancing global search and local search, is the inertia weight.

The inertia weight, *w*, is improved to be linearly decreasing and is changed from a constant to a variable.
(8)$$w\;=\;w_{\max}\;-\;\frac{t\;*\;(w_{\max}\;-\;w_{\min})}{t_{\max}}$$
where *t* is used to represent the current iteration, *t*_max_ is the total number of iterations predetermined by the algorithm.

The Bouc–Wen model usually represents the relationship between the output signal and the input signal by constructing a differential equation. As a typical differential equation-type hysteresis model, it is more intuitive and simpler than other typical subclass hysteresis models [20,21].

The general classical Bouc-Wen model is employed to improve the control performance of the piezo-actuated stage when the input frequency is within 1 Hz.
(9)$$\begin{array}{l} x\;=\;k_{1c}u\;+\;k_{3}{}_{c}h\\ \dot{h}\;=\;\alpha_{c}\dot{u}\;-\;\beta_{c}\;|\dot{u}|\;|h|\;h\;-\;\gamma_{c}\dot{u}\;|h{|}^{2} \end{array}$$

If the input frequency is higher than 1 Hz, the proposed mathematical expression in this paper is as follows:(10)$$\begin{array}{l} x\;=\;k_{1}e^{-\;k_{2}\dot{u}}u\;+\;k_{3}h\\ \dot{h}\;=\;\eta \;(\alpha e^{\varepsilon |\dot{u}|}\dot{u}\;-\;\beta \;|\dot{u}|h\;-\;\gamma \dot{u}|h|\;+\;\lambda |\dot{u}|\;) \end{array}$$
where $\dot{h}$ is the derivative of the hysteresis $\dot{h}$. The parameters, *u* and $\dot{u}$, are respectively the input voltage and its derivative. The coefficients *α*, *β*, *γ*, *ε* are the parameters of the modified Bouc-Wen model, *λ* is control factor related to the bias input signal, and *η* is the adjustable factor for the hysteresis of the different input signal. From the Figure 3, the hysteresis curved reveals the similar rate-independent characteristics when the frequency of the input signal is less than 1 Hz. Therefore, $e^{-\;k_{2}\dot{u}}$, $e^{\varepsilon \;|\dot{u}|}$ will converge to fixed constants under the low frequency. But the hysteresis phenomenon of this stage is regarded as rate-dependent when the input frequency is more than 1 Hz. So the parameters $e^{-\;k_{2}\dot{u}}$
$e^{\varepsilon \;|\dot{u}|}$ will vary with different frequency [3].

The objective function expressed by the root-mean-square error is given as:(11)$$F\;(k_{1},\;k_{2},\;k_{3},\;\alpha ,\;\beta ,\;\gamma ,\;\lambda ,\;\eta ,\;\varepsilon )\;=\;\sqrt{\frac{1}{N}\sum_{1}^{N}\;{\left[x_{e}\;(k)\;-\;x\;(k)\;\right]}^{2}}$$

A sinusoidal voltage signal, $u\;=\;4.58\;+\;4.58\sin\;(2\pi ft\;-\;\frac{\pi }{2})\;(f\;=\;1)$, is used to this stage to identify the parameters of Equation (9). The identification results are listed as: *k*_1c_ = 1.86, *k*_3c_ = −0.75, *α*_c_ = 1.14, *β*_c_ = 2.49, *γ*_c_= −2.08.

A set of asymmetric sinusoidal voltage signals, $u\;=\;8.68\;+\;7.96\sin(2\pi ft\;-\;\frac{\pi }{2})\;(f\;=\;1,\;2,\;5)$, are utilized to stimulate the piezo-actuated stage. The comparisons of experimental and simulation results based on a modified Bouc-Wen model are shown in Figure 4.

The optimal parameters of the Bouc–Wen model have been identified through a set of experimental data. The identified parameters of the proposed Bouc-Wen model are listed in Table 1.

## 4. Control

The actual input–output relationship of the piezo-actuated stage between the input voltage and the output displacement is nonlinear because of the hysteresis of the piezoelectric actuator. The hysteresis loop can be decomposed into a linear element *d*(*t*) and a non-linear element *H*(*t*) [4,22]. Therefore, the output displacement of this piezo-actuated stage can be given by
(12)$$\begin{array}{l} x(t)\;=\;d(t)\;+\;H(t)\\ d(t)\;=k_{l}u(t)\\ H(t)\;=\;k_{H}h(t) \end{array}$$
where *k*_l_ and *k_H_* is proportional gain related to the input voltage and the hysteresis, respectively.

The optimal parameters of the Bouc–Wen model have been identified, so the hysteresis of the piezo-actuated stage can be estimated. The hysteresis behavior of this stage is suppressed by designing a feedforward controller based on an inverse hysteresis model. If the input reference displacement is *x_d_(t)*, then the reference input voltage can be given by:(13)$$u\;=\;\frac{x_{d}(t)}{k_{F}}$$
where *k_F_* represents the relationship describing the ratio of the displacement voltage.

Due to the existing of the hysteresis phenomenon, the actual displacement output of this stage based on open-loop control is given as:(14)$$x(t)\;=\;k_{F}u\;+\;k_{H}h(t)$$

If the hysteresis nonlinearity can be estimated by the Bouc–Wen model, the feedforward compensator based on the Bouc-Wen model can be designed as follows:(15)$$u_{FF}\;=\;\frac{x_{d}(t)\;-\;k_{H}\hat{h}}{k_{F}}$$
where $\hat{h}(t)$ represents the estimated hysteresis of the piezo-actuated stage based on the Bouc–Wen model. In practical applications, due to the existence of model errors and external disturbances, it is difficult to obtain satisfactory control performance by only using the feedforward control, so the feedback control is particularly important to achieve precision performance.

The composite control with feedforward control and feedback control is used to operate the piezo-actuated stage. In the feedforward control, the hysteresis compensation voltage *u*_FF_ is obtained according to the inverse compensator of the identified Bouc–Wen model. In the feedback loop, the PID control algorithm is employed to adjust the input voltage *u*_FB_ of the closed-loop control.
(16)$$u(t)\;=\;u_{FF}(t)\;+\;u_{FB}(t)$$

According to the PID controller in the Programmable Multi Axis Controller (PMAC) card from Delta Tau Data Systems, Inc., the voltage of feedback control is expressed as:(17)$$u_{FB}(k)\;=\;K_{p}\;*\;e(k)\;+\;K_{I}\sum_{i=1}^{k}e(i)\;-\;K_{D}[x(k)\;-\;x(k\;-\;1)]$$

In the practical application, the sensible selection of the gains *K*_P_, *K*_I_ and *K*_D_ can make the system obtain the desired performance, such as fast convergence, less overshoot, and small error [23]. In the compound control, the feedforward control is employed to reduce the non-linear hysteresis effect of the piezoelectric actuator, while the fuzzy PID feedback control can be proposed to achieve stabilization and disturbance compensation. The block diagram of its compound control is described in Figure 5.

The voltage based on the feedforward compensator and the fuzzy PID feedback control can be expressed as:(18)$$\begin{array}{l} u(k)\;=\;u_{FF}(k)\;+\;u_{FB}(k)\;=\;H^{-1}(x_{d}(k))\;+\;K_{p}(k)\;*\;e(k)\;+\;K_{I}(k)\sum_{i=1}^{k}e(i)\;-\;K_{D}(k)[x(k)\;-\;x(k-1)]\\ H^{-1}(x_{d}(k))\;=\;\frac{x_{d}(t)\;-\;k_{H}\hat{h}}{k_{F}} \end{array}$$

When the input frequency is less than 1 Hz, then:(19)$$\begin{array}{l} k_{F}\;=\;k_{1c}\\ \dot{\hat{h}}\;=\;\alpha_{c}\dot{u}\;-\beta_{c}|\dot{u}||\dot{\hat{h}}|\dot{\hat{h}}\;-\;\gamma_{c}\dot{u}|\dot{\hat{h}}{|}^{2} \end{array}$$

But if the input frequency is higher than 1 Hz:(20)$$\begin{array}{l} k_{F}\;=\;k_{1}e^{-k_{2}\dot{u}}\\ \dot{\hat{h}}\;=\;\eta (\alpha e^{\varepsilon |\dot{u}|}\dot{u}\;-\;\beta |\dot{u}|\dot{\hat{h}}\;-\;\gamma \dot{u}|\dot{\hat{h}}|\;+\;\lambda |\dot{u}|) \end{array}$$

Two variables as input parameters, displacement error *e* and the differential of error *ec*, are proposed to tune the gains of PID by using fuzzy rules. The form of the gains of fuzzy PID is expressed as follows:(21)$$\begin{array}{l} k_{P}(k)\;=\;k_{P0}\;+\;\Delta k_{P}(k)\\ k_{I}(k)\;=\;k_{I0}\;+\;\Delta k_{I}(k) \end{array}$$

The linguistic variables of the fuzzy controller are expressed as follows: negative large for NB, negative medium for NM, negative small for NS, zero for ZO, positive small is PS, positive medium is PM, and positive large is PB. These fuzzy membership functions are defined as the triangle function. Two inputs: error, i.e, *e*, and derivative of error, i.e, *ec*, are employed as the inputs of fuzzy controller.

The proportional gain *K*_P_ is too small, the response of the system is slow, and also the accuracy is low. The integral *K*_I_ can eliminate the error of the control system, but it will produce overshoot when its value is too large. The adjustment rules of the parameters shown in Table 2 are used to design the parameters Δ*K*_P_, Δ*K*_I_.

The derivative *K*_D_ can improve the dynamic characteristics of the control system and suppress the error’s change. However, when *K*_D_ is too high, the settling time will increase, and also the robustness of the system will be reduced [24,25]. In practical applications, the effect of the derivative gain held steady under some range. Therefore, the value of *K*_D_ was finally determined to be 120.

## 5. Experimental Results and Discussion

The experimental setup is built and shown in Figure 6. This setup is composed of the PA (P-845.20, Physik Instrumente, Karlsruhe, Germany), the amplifier driver (E-625, Physik Instrumente, Karlsruhe, Germany), the motion control card (Turbo PMAC), a grating encoder of Heidenhain and the computer. A PA is adapted to actuate the flexure hinges guide mechanism. Turbo PMAC, a motion control card, is employed to generate excitation voltage signals (0–10 V). By using the amplifier driver E-625, the excitation voltage signals can be amplified by 10 times. The output displacement is measured by grating encoder.

The control comparison is demonstrated by using PID control and fuzzy PID control with the inverse hysteresis model. The experimental result is shown in the Figure. The blue dotted line in the Figure represents the experimental curve of PID control, and the red short dotted line reveals the experimental result of fuzzy PID control with inverse hysteresis model.

The experimental results of the piezo-actuated stage using different controllers under 6 μm–0.5 Hz and 1 Hz sinusoidal signal are shown in Figure 7 and Figure 8. The control errors of the two control strategies are demonstrated in Figure 7 and Figure 8b. Observing Figure 7, the maximum absolute errors based on PID and inverse model compensator with fuzzy PID are 0.182 μm and 0.057 μm, respectively. From Figure 8, the maximum absolute errors based on PID and inverse model compensator with fuzzy PID are 0.318 μm and 0.045 μm, respectively. The proposed controller can reduce the error by about 85%. Compared to the input–output characteristics in the (c) and (d) part of Figure 7 and Figure 8, the initial hysteresis of the piezo-actuated stage has been removed when utilizing the proposed controller.

The system responses of 5 Hz input within amplitude 4 μm and 10 Hz sinusoidal signal within amplitude 6 μm are illustrated in Figure 9 and Figure 10, respectively. Observing Figure 9, the maximum absolute error based on PID is 0.429 μm, and the maximum error of the compound controller is 0.078 μm. The error is reduced by nearly 82%. In Figure 10, the max measured error under PID control and the compound control are 2.27 μm and 0.179 μm, respectively. The error is reduced by nearly 92%.

To further verify the accuracy and robustness of identified model and the proposed controller, a composite signal $x\;=\;5*\sin(2\pi f_{1}t)\;+\;2*\sin(2\pi f_{2}t)\;f_{1}\;=\;1,\;f_{2}\;=\;5$ was applied. The output displacement results are shown in Figure 11. From the experimental results, the maximum tracking error using PID is 6.10 μm, while the maximum error based on the compound controller is about 0.36 μm. The error is reduced by nearly 94%. Compared to the control results, the proposed controller can improve the control error.

## 6. Conclusions

This paper presents the study of the identification of a model Bouc–Wen model using a modified PSO and a compound control. The modified Bouc–Wen model is divided into a rate-independent type and rate-dependent type according to the input frequency. The Bouc–Wen is supposed to be the rate-independent model when the input frequency is under 1 Hz. However, if the input frequency is larger than 1 Hz, the Bouc–Wen model should be treated as a rate-dependent model. The feedforward compensator is constructed by an inverse Bouc–Wen model. The gains of the PID control are adjusted by fuzzy rules. For the frequency superposition signal, the error is reduced by about 94%. Compared to the control results, the proposed controller can improve the control error.

## Figures and Tables

**Figure 1 micromachines-10-00861-f001:**
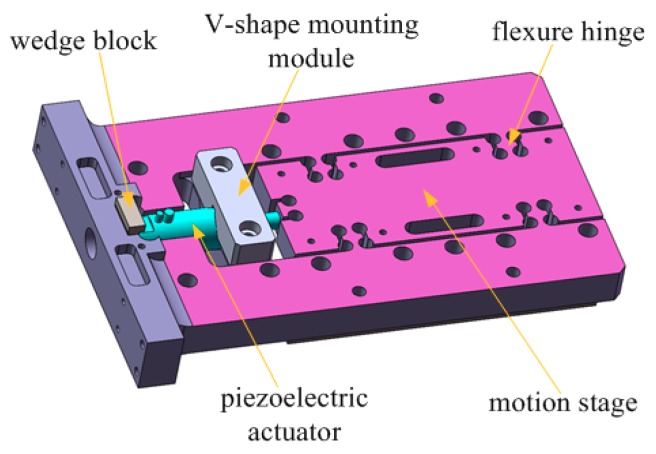
The mechanical structure of the piezo-actuated stage.

**Figure 2 micromachines-10-00861-f002:**
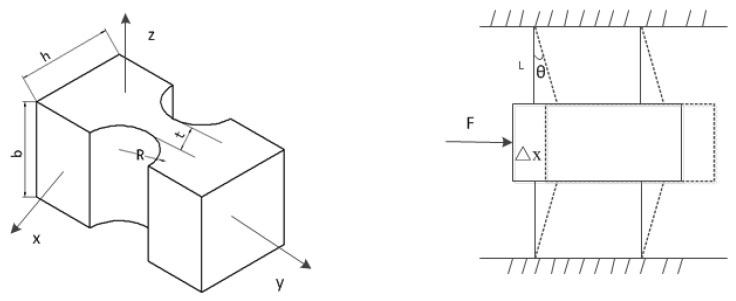
The model of the right circular flexure hinges and schematic diagram of the guide mechanism.

**Figure 3 micromachines-10-00861-f003:**
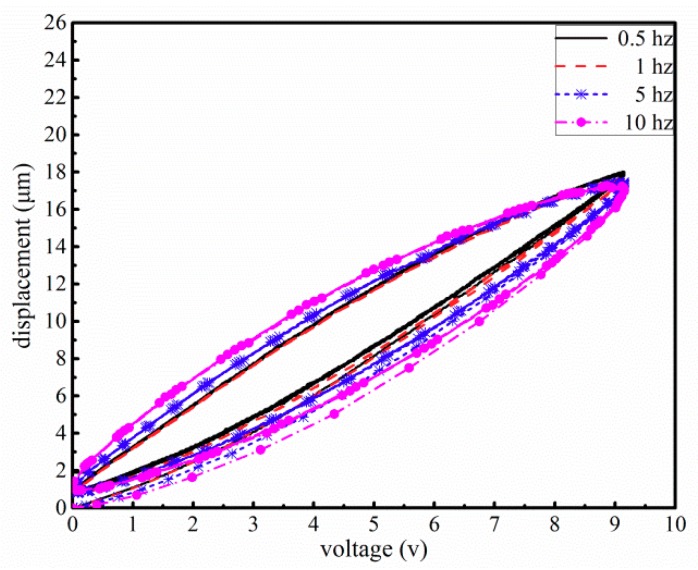
The hysteresis curves with input voltage signals of different frequencies.

**Figure 4 micromachines-10-00861-f004:**
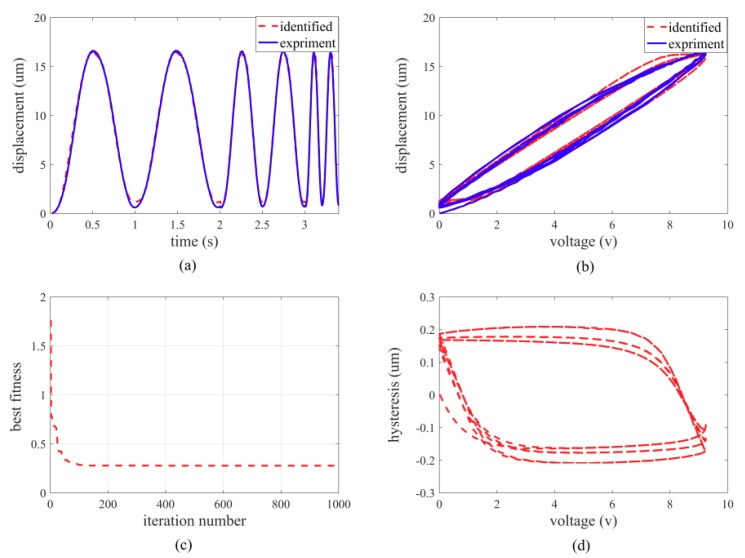
Comparisons of experimental and simulation results based on the Bouc-Wen model. (**a**) displacement; (**b**) hysteresis loop; (**c**) iteration curve; (**d**) non-linear component.

**Figure 5 micromachines-10-00861-f005:**
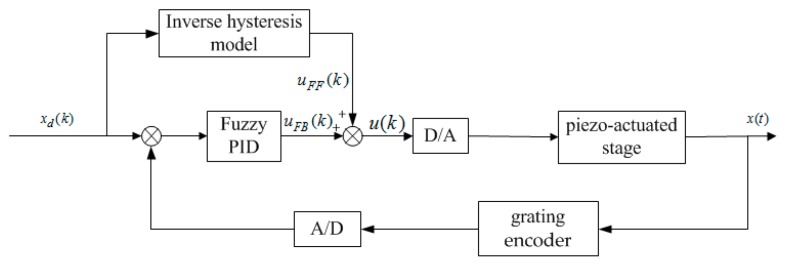
The block diagram of the fuzzy proportional-integral-derivative (PID) controller with inverse model feedforward compensator.

**Figure 6 micromachines-10-00861-f006:**
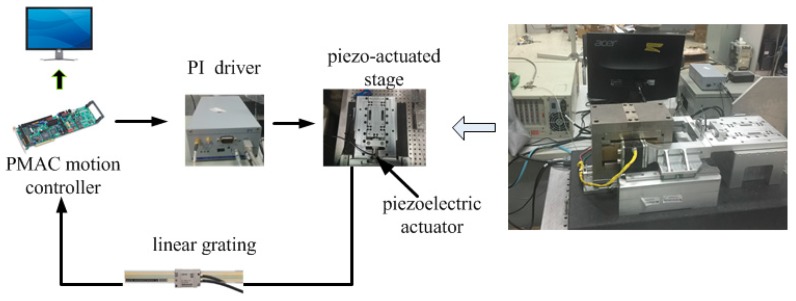
The experimental setup.

**Figure 7 micromachines-10-00861-f007:**
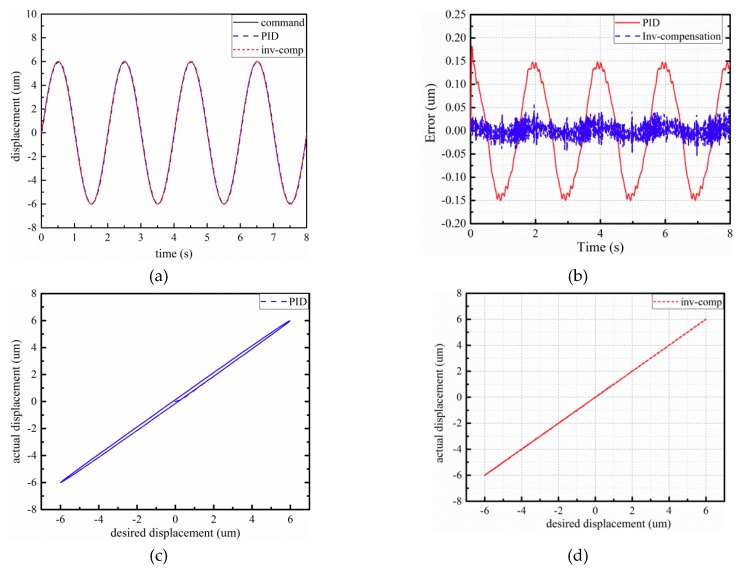
Experimental results of 6 μm–0.5 Hz sinusoidal signal: (**a**) displacement response; (**b**) tracking error; (**c**) input–output characteristics with PID controller; (**d**) input–output characteristics with the proposed controller.

**Figure 8 micromachines-10-00861-f008:**
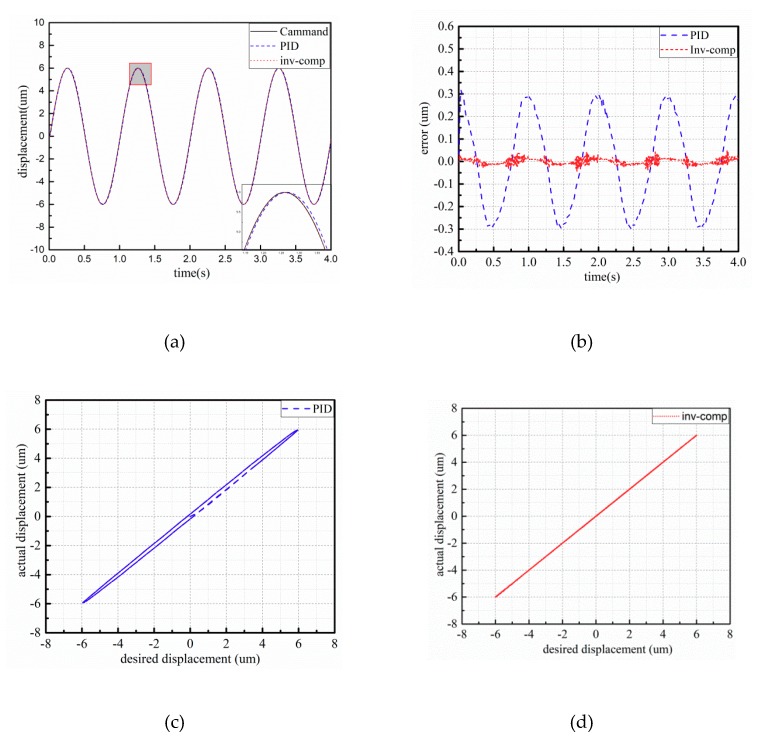
Experimental results of 6 μm–1 Hz sinusoidal signal: (**a**) displacement response; (**b**) tracking error; (**c**) input–output characteristics with PID controller; (**d**) input–output characteristics with the proposed controller.

**Figure 9 micromachines-10-00861-f009:**
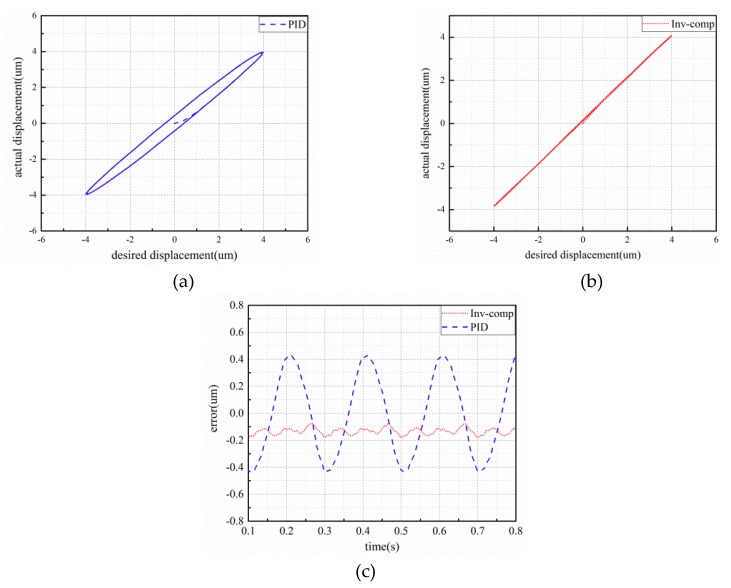
Experimental results of 4 μm–5 Hz sinusoidal signal: (**a**) input–output characteristics with PID controller; (**b**) input–output characteristics with the proposed controller; (**c**) tracking error.

**Figure 10 micromachines-10-00861-f010:**
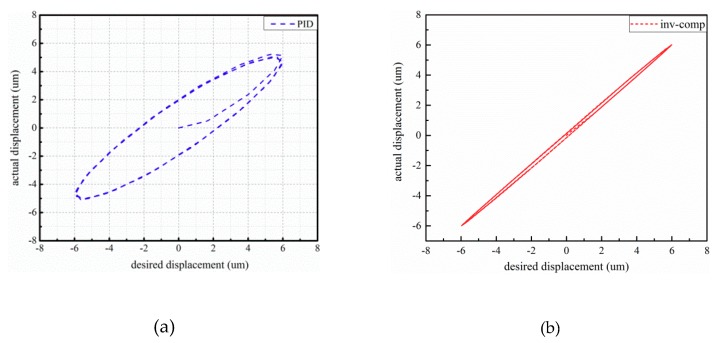
Experimental results of 6 μm-10 Hz sinusoidal signal: (**a**) input–output characteristics with PID controller; (**b**) input–output characteristics with the proposed controller; (**c**) tracking error.

**Figure 11 micromachines-10-00861-f011:**
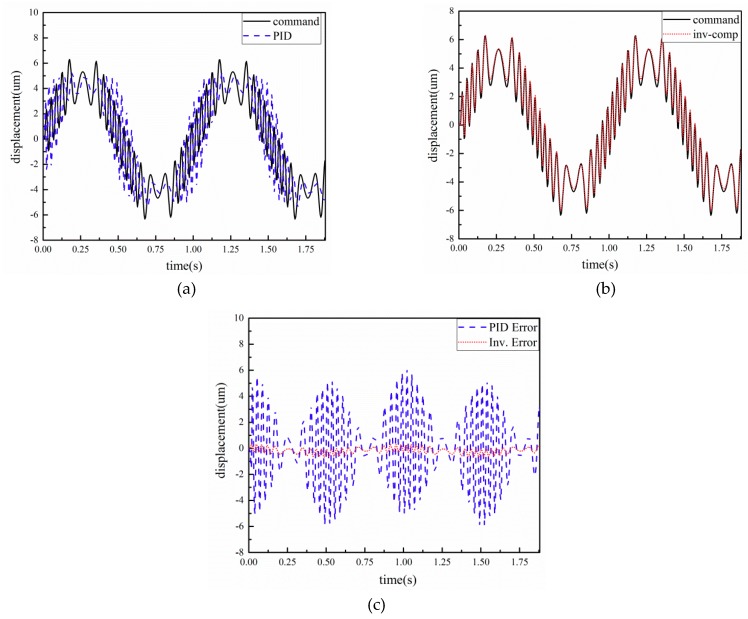
Experimental results of $x\;=\;5*\sin(2\pi f_{1}t)\;+\;2*\sin(2\pi f_{2}t)\;f_{1}\;=\;1,\;f_{2}\;=\;5$: (**a**) displacement response using PID controller; (**b**) displacement response using the proposed controller; (**c**) tracking error.

**Table 1 micromachines-10-00861-t001:** The identified parameters of the hysteresis model.

Parameters	Value
*k* _1_	1.86
*k* _2_	0.11
*k* _3_	6.82
*α*	−2.27
*β*	7.19
*γ*	−6.29
*ε*	0.35
*λ*	0.0054
*η*	0.089

**Table 2 micromachines-10-00861-t002:** Adjustment rules table of Δ*K*_P_/Δ*K*_I_.

	*ec*	NB	NM	NS	ZO	PS	PM	PB
*e*								
**NB**		PB/NB	PB/NB	PM/NM	PM/NM	PS/NS	ZO/ZO	ZO/ZO
**NM**		PB/NB	PB/NB	PM/NM	PS/NS	PS/NS	ZO/ZO	NS/ZO
**NS**		PM/NB	PM/NM	PS/NS	PS/NS	ZO/ZO	NS/PS	NS/PS
**ZO**		PM/NM	PM/NM	PS/NS	ZO/ZO	NS/PS	NM/PM	NM/PM
**PS**		PS/NM	PS/NS	ZO/ZO	NS/PS	NS/PS	NM/PM	NM/PB
**PM**		PS/ZO	ZO/ZO	NS/PS	NM/PS	NM/PM	NM/PB	NB/PB
**PB**		ZO/ZO	ZO/ZO	NM/PS	NM/PM	NM/PM	NB/PB	NB/PB

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
