# Peer review of "A Compound Control Based on the Piezo-Actuated Stage with Bouc–Wen Model"

_micromachines, 2019, doi:10.3390/mi10120861_

Round 1

Reviewer 1 Report

Comments to Author:
Title: A compound control based on the piezo‐actuated stage with Bouc‐Wen model
Overview and general recommendation:
In the proposed manuscript, the authors describe a modified Bouc‐Wen model used to compensate the
positioning errors due to hysteresis nonlinearity in piezo‐actuated stages. The proposed model is divided into
two categories according to the different effects of the input frequency: rate‐independent type and ratedependent
type. Experimental results indicate that the proposed control strategy, based on a fuzzy
proportional‐integral‐derivative combined with the feedforward compensator, can compensate the
hysteresis phenomenon. In particular, the obtained results are very interesting since the absolute errors are
significantly reduced respect to those achieved using a PID control.
In my opinion, however, the manuscript is not suitable for publication in Micromachines in the present form
and minor issues have to be addressed.
Minor comments:
(a) The acronym “PZT” is used to indicate the “piezoelectric actuator”. Anyway, PZT is commonly used
to indicate “Lead zirconate titanate”. The authors should use the acronym PZT only for “Lead
zirconate titanate” to avoid misunderstandings.
(b) Section 2: “Mechanical Structure of Piezo‐actuated Stage”.
The mechanical structure of the piezo‐actuated stage could be better described. In fact, it is not clear
if the schematic drawing of figure 1 is the actual stage used in the experiment. In my opinion, the
description of the piezo‐actuated stage (rows: 96‐102) including also the number of the circular
flexure hinges, should be reported before of the motion displacement description provided in
equation 1‐3.
(c) How the authors obtained the results reported in Fig.3? Are these results obtained by using the
experimental set‐up described in section 5?
(d) Row 136: the inertia weight w is introduced after equation (6). Anyway, w is not indicated in eq. (6),
but only in eq. (7). This point need to be better explained.
(e) Section 5. The piezo‐actuated stage based on circular flexure hinges is, obviously, the same of that
described in section 2. Anyway, it would be interesting for readers to know if this stage is a
commercially available system or not.
(f) Row 300. The authors used a frequency superposition signal to further verify the performance of the
model. Why? Is the motivation to model a noise signal? This aspect could be explain in the text.

Reviewer 2 Report

This paper could published to micromachines, but I recommend that authors have to check below.

Usually, PZT means Pb(Zr,Ti)O3, Lead zirconate titanate, or piezoelectric transducers. The paper express PZT as the piezoelectric actuator. That can be confusing. line 97; references have to be needed which authors can find the real parameters ( E = 71 GPa, density = 2800 kg/m3, etc). line 146 what is t*? expression of unit : 1Hz→1 Hz, 2800kg/m3→ 2800 kg/m3, 3.5mm → 3.5 mm; Authors have to check all the expressions. in Fig.3; 0.5hz, 1hz, etc → 0.5 Hz, 1 Hz, etc 165 line; eror → error line 148 - 150;  Reference have to be needed.

Round 2

Reviewer 1 Report

Dear Authors,

In my opinion, the revised version of the manuscript is suitable for publication on Micromachines.

Reviewer 2 Report

The authors have addressed my comments and the manuscript is now suitable for publication.